# Fenton Process for Treating Acrylic Manufacturing Wastewater: Parameter Optimization, Performance Evaluation, Degradation Mechanism

Zhiwei Lin, Chunhui Zhang *, Peidong Su, Wenjing Lu, Zhao Zhang, Xinling Wang and Wanyue Hu

College of Chemistry and Environmental Engineering, China University of Mining and Technology (Beijing), Beijing 100083, China
* Correspondence: ZCHcumtb@hotmail.com; Tel.: +86-139-1017-6209

**Abstract:** Acrylic manufacturing wastewater is characterized by high toxicity, poor biodegradability, high chemical oxygen demand (COD) and ammonia nitrogen. Herein, we exploited traditional Fenton technology to treat acrylic fiber manufacturing wastewater. The impacts of key operating variables including the initial concentration of $H_2O_2$ ($C_{H2O2}$), the initial concentration of $Fe^{2+}$ ($D_{Fe2+}$), and solution pH (pH) on the COD removal rate ($R_{COD}$) were explored and the treatment process was optimized by Response Surface Methodology (RSM). The results indicated that the optimum parameters are determined as pH 3.0, 7.44 mmol/L of $Fe^{2+}$ and 60.90 mmol/L of $H_2O_2$ during Fenton process. For the actual acrylic manufacturing wastewater treatment shows that the removal rates for COD, TOC, $NH_4^+$-N and TN are 61.45%~66.51%, 67.82%~70.99%, 55.67%~60.97% and 56.45%~61.03%, respectively. It can meet the textile dyeing and finishing industry water pollutant discharge standard (GB4287-2012). During the Fenton reaction, the effective degradation and removal of organic matter is mainly achieved by $HO\bullet$ oxidation, supplemented by flocculation and sedimentation of $Fe^{3+}$ complexes. This study will provide useful implications in the process parameters for the practical application of Fenton method in acrylic acid production wastewater.

**Keywords:** acrylic fiber manufacturing wastewater; Fenton reaction; advanced oxidation method; degradation mechanism

## 1. Introduction

Acrylic fibers are one kinds of the significant manufacturing raw materials for the textile industry [1]. During the manufacture of acrylic fibers, a large volume of wastewater is inevitably generated. Owing to its high toxicity, poor biodegradability, high COD, and ammonia nitrogen, acrylic fiber manufacturing wastewater has been recognized as one of the problematic organic wastewaters [2]. In general, conventional biological methods have been used in the degradation of organic matters in acrylic fiber manufacturing wastewater. However, the effluent water quality treated by conventional biological processes such as A/O and $A^2$/O alone usually cannot meet the national discharge standards [3]. Therefore, it is imperative to develop stable, effective, economical combined processes to treat acrylic manufacturing wastewater.

At present, the main approaches for the treatment of acrylic fiber manufacturing wastewater include but not limit to bio-enhanced treatment [4,5], membrane technology [6,7], advanced oxidation processes (AOP) [8,9], and coupled treatment processes [10]. C. Gong et al. [11] used an electro-coagulation (EC) process to pretreat acrylic fiber manufacturing wastewater. Under optimal conditions, the removal rate of total organic carbon was 44%, and the (Biochemical Oxygen Demand) $BOD_5$/COD ratio was increased to 0.35. J. Wei et al. [12] combined Fenton-sequencing batch membrane bioreactor process for the treatment of acrylic fiber manufacturing wastewater. After Fenton oxidation treatment, the COD removal rate was 47.0%, and the ratio of $BOD_5$/COD increased from 0.35 to

0.69. Microbubble-ozonation was used to treat refractory wet-spun acrylic fiber manufacturing wastewater with 42%, 21%, and 42% removal rates of COD, $NH_3$-N, and $UV_{254}$, respectively [13]. The degradation of refractory organics such as alkane aromatic compounds in wastewater was enhanced by microbubble ozone treatment, which resulted in improved biodegradability of the wastewater. J. Wei et al. [14] utilized the Fered-Fenton process to treat acrylic fiber manufacturing wastewater, resulting in an increase in the $BOD_5$/COD ratio from 0.29 to over 0.68 after 180 min of treatment in which Ti was used as the cathode and $RuO_2$/Ti as the anode. T. Zheng et al. [15] fabricated a three-dimensional electrochemical oxidation reactor to treat wet-spun acrylic fiber manufacturing wastewater, which significantly elevated the treatment efficiency of COD, $NH_3$-N, TOC, and $UV_{254}$ by 44.5%, 38.8%, 27.2%, and 10.9%, respectively. X. Xu and Z. Shao et al. [16] developed highly efficient peroxy monosulfate activated catalysts (e.g., $LaSrCo_{0.8}Fe_{0.2}O_4$) that showed excellent performance in catalytic advanced oxidation applications of difficult to degrade organic pollutants. In terms of this, the enhanced pretreatment of acrylic fiber wastewater using advanced oxidation processes can achieve satisfactory treatment results.

Fenton advanced oxidation process, which is based on the generation of hydroxyl radicals (HO•), has the advantages of high oxidation and strong electron affinity [17]. The Fenton process can decompose refractory organic matter, and its by-product, $Fe^{3+}$, have a favorable flocculation effect, which can simultaneously complete the degradation-coagulation-precipitation removal of organic matter. Therefore, the Fenton method is considered an effective method for treating refractory organic wastewater [18]. However, there are many intermediate products and side reactions in the Fenton reaction system. Once the reaction conditions are not adequately controlled, the utilization rate of $H_2O_2$ will decrease, resulting in insufficient mineralization of organic matter and a severe waste of raw materials [19]. Therefore, during the application of the Fenton method, the use of reagents, the conversion trend of $H_2O_2$, and the influence of operating parameters on the reaction system should be investigated to achieve the efficient utilization of the Fenton process and the economic degradation of pollutants. As mentioned above, this study adopts the response surface methodology to explore the effect of factors such as the concentration of $H_2O_2$, the amount of $Fe^{2+}$ added, and the initial pH value on the degradation of organic matter. In addition, we discussed the interaction between various factors, the removal effect of refractory organic matter, and the mechanism of the reaction process. This research provides technical reference and a theoretical basis for applying the Fenton catalytic oxidation method in the treatment of acrylic fiber manufacturing wastewater.

## 2. Materials and Methods

### 2.1. Wastewater

The experimental acrylic fiber wastewater was obtained from the effluent of the secondary sedimentation tank after A/O process treatment in a sewage treatment plant in Jilin Province. The experimental wastewater was pretreated with concentrated sulfuric acid within 12 h after sampling. The main purpose of pretreatment of acrylic acid wastewater with concentrated sulfuric acid was to lower the pH of the solution to about 2. The storage time of acrylic wastewater was prolonged by lowering the pH of the water sample to slow down the microbial activity and inhibit the hydrolysis of ammonia-containing compounds. It is worth noting that the water samples must be used within 48 h. The physical and chemical indicators of wastewater are shown in Table 1.

**Table 1.** Water quality indexes of water samples.

| Parameter | COD mg·L$^{-1}$ | BOD$_5$ mg·L$^{-1}$ | TOC mg·L$^{-1}$ | NH$_3$-N mg·L$^{-1}$ | TN mg·L$^{-1}$ | BOD$_5$/COD | pH |
|---|---|---|---|---|---|---|---|
| Amount | 249–270 | 8–10 | 90–100 | 60–66 | 78–95 | 0.030–0.040 | 5.4–5.8 |

### 2.2. Chemicals

All chemicals are of analytical grade. NaOH and $H_2SO_4$ (98%) were purchased from Beijing Chemical Reagent Co., Beijing, China. $FeSO_4 \cdot 7H_2O$ and $H_2O_2$ (30%) were purchased from Beijing Lanyi Chemical Co., Beijing, China.

### 2.3. Experimental Method

Since the degradation efficiency of COD by Fenton process was usually affected by the initial pH of the solution, the $H_2O_2/FeSO_4$ ratio, the concentration of $H_2O_2$, $FeSO_4$, reaction time, and temperature [20,21]. The initial concentrations of $Fe^{2+}$ and $H_2O_2$ as well as the pH determined the amount of HO• production in the Fenton reaction, which was the active substance for the direct oxidation of COD [2,22–24]. The initial concentration of $H_2O_2$ (code A), the initial concentration of $Fe^{2+}$ (code B), and the pH value (code C) was chosen as the primary factors. Firstly, single-factor experiments were conducted to explore the effect of COD degradation under different conditions. After that, response surface methodology was adopted to optimize conditions and analyze the interaction between factors to optimize the Fenton process. Taking the removal rate of COD as the response value, a three-factor and three-level Box-Behnken response surface design was carried out. The experimental factors, level codes, design schemes, and response values are shown in Table 2.

**Table 2.** Coded levels and corresponding values for test factors in RSM experimental design.

| Run | A: [$H_2O_2$] | | B: [$Fe^{2+}$] | | C: pH | | Response, R (%) |
| :---: | :---: | :---: | :---: | :---: | :---: | :---: | :---: |
| | Coded Level | Corresponding Value (mmol·L$^{-1}$) | Coded Level | Corresponding Value (mmol·L$^{-1}$) | Coded Level | - | |
| S1 | −1 | 56.25 | −1 | 4.16 | 0 | 3 | 51.92 ± 1.25 |
| S2 | 1 | 93.75 | −1 | 4.16 | 0 | 3 | 43.73 ± 1.02 |
| S3 | −1 | 56.25 | 1 | 9.38 | 0 | 3 | 54.91 ± 1.28 |
| S4 | 1 | 93.75 | 1 | 9.38 | 0 | 3 | 45.57 ± 1.06 |
| S5 | −1 | 56.25 | 0 | 6.25 | −1 | 2 | 44.72 ± 0.98 |
| S6 | 1 | 93.75 | 0 | 6.25 | −1 | 2 | 39.66 ± 0.89 |
| S7 | −1 | 56.25 | 0 | 6.25 | 1 | 4 | 50.63 ± 1.05 |
| S8 | 1 | 93.75 | 0 | 6.25 | 1 | 4 | 42.17 ± 1.07 |
| S9 | 0 | 75.00 | −1 | 4.16 | −1 | 2 | 34.95 ± 0.67 |
| S10 | 0 | 75.00 | 1 | 9.38 | −1 | 2 | 49.71 ± 1.09 |
| S11 | 0 | 75.00 | −1 | 4.16 | 1 | 4 | 38.16 ± 0.88 |
| S12 | 0 | 75.00 | 1 | 9.38 | 1 | 4 | 41.42 ± 0.87 |
| S13 | 0 | 75.00 | 0 | 6.25 | 0 | 3 | 57.04 ± 1.35 |
| S14 | 0 | 75.00 | 0 | 6.25 | 0 | 3 | 57.88 ± 1.39 |
| S15 | 0 | 75.00 | 0 | 6.25 | 0 | 3 | 56.89 ± 1.38 |
| S16 | 0 | 75.00 | 0 | 6.25 | 0 | 3 | 57.28 ± 1.42 |
| S17 | 0 | 75.00 | 0 | 6.25 | 0 | 3 | 57.54 ± 1.36 |

The experimental setup and procedure are shown in Figure 1. Firstly, 300.0 mL of acrylic fiber wastewater was added to the 500.0 mL beaker. The pH adjustment was achieved by titrating 0.1 or 1 mol/L of $H_2SO_4$ (or NaOH) solution under the condition of stirring intensity around 140 rpm. In which the pH of the solution was monitored by (LEICI PHB-4) portable pH meter to achieve precise regulation of the solution pH. Secondly, $FeSO_4 \cdot 7H_2O$ was added to the solution and stirred at 140.0 rpm. In addition, $H_2O_2$ (30% in mass fraction) was added to the solution to initiate the Fenton oxidation reaction. Finally, samples were taken at intervals to test the concentrations of $Fe^{2+}$, COD, and $H_2O_2$.

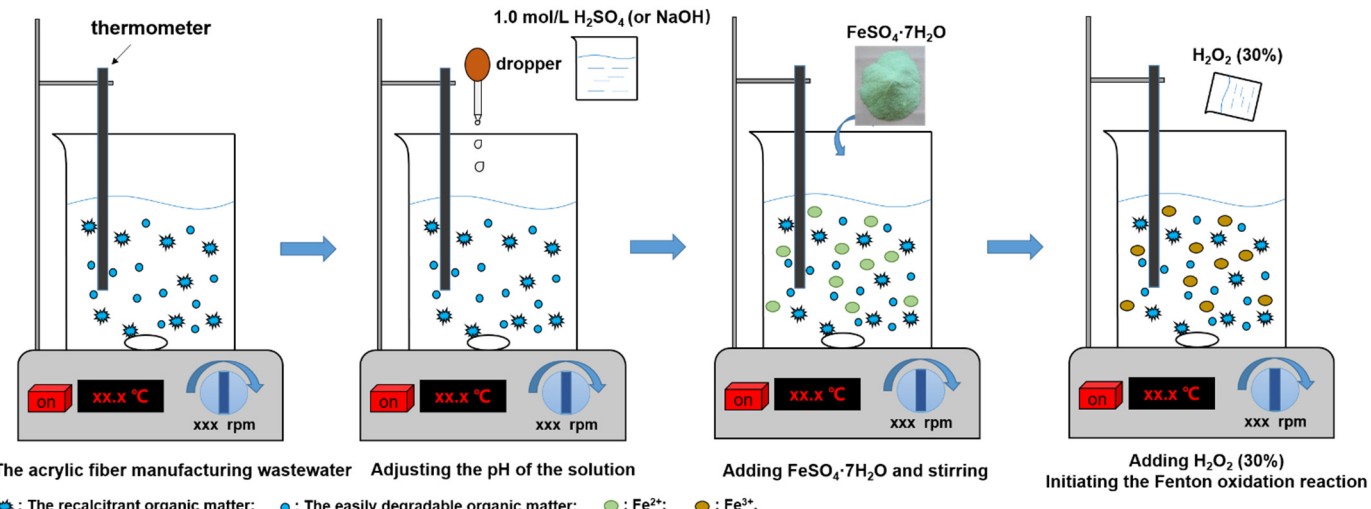

**Figure 1.** Fenton oxidation experiment.

*2.4. Analytical Methods*

The method of COD detection refers to the standard method "Water quality-Determination of the chemical oxygen demand-Dichromate method, HJ 828-2017" published by the Ministry of Ecology and Environment of the People's Republic of China [25]. The detection method of ammonia nitrogen refers to the standard method "Water quali-ty-Determination of ammonia nitrogen-Nessler's reagent spectrophotometry, HJ535-2009" published by the Ministry of Ecology and Environment of the People's Republic of China [26]. The determination method of iron content refers to the standard method "Water quality-Determination of Iron-phenanthroline spectrophotometry, HJ/T 345-2007" published by the Ministry of Ecology and Environment of the People's Republic of China [27]. The concentration of $H_2O_2$ was determined spectrophotometrically with potassium titanium oxalate. 5 mL of water sample was added to a 25 mL cuvette. 5 mL of potassium titanium oxalate solution (0.02 mol/L) was added. Deionized water was used to dilute to the scale. After standing for 8 min, the absorbance was determined at 385 nm. Deionized water was used as a reference during the determination [28,29].

GC-MS (Gas chromatograph-mass spectrometer) was used to detect the type (or concentration) of organics. The supernatant before and after the Fenton reaction was first filtered through a 0.45 mm membrane. Then the aqueous samples were then subjected to solid phase extraction, elution, dehydration and concentration procedures to complete the aqueous sample pretreatment. Samples were analyzed by GC-MS (GC (7890)-MS (5975), Agilent Technologies Inc.) equipped with a separation column (DB-5MS 30 m × 0.25 mm × 0.25 μm) with column template at 40 °C for 2 min. The inlet temperature was 290 °C for 4 min. Carrier gas conditions: high-purity nitrogen with a flow rate of 1.0 mL/min. MS conditions: ion source temperature 280 °C, interface temperature 280 °C, solvent delay time 5.0 min.

## 3. Results and Discussion

*3.1. Examination of Main Factors That Affect the Removal of COD*

In the Fenton reaction system, the initial concentration of $Fe^{2+}$ and $H_2O_2$ as well as the pH value play a crucial role in the degradation of organic pollutants. [30]. Since they determine the amount of HO• production in the Fenton reaction, which is the active species for the direct oxidation of COD. Typically, the initial concentration of $Fe^{2+}$ and pH was set to be 2~10 mmol/L and 3.0, respectively [31].

### 3.1.1. Initial Concentration of $H_2O_2$

Under this condition: (i) the initial concentration of $Fe^{2+}$ was 6.25 mmol/L; (ii) the initial pH was 3.0; The effect of the initial concentration of $H_2O_2$ on the COD removal rate of acrylic fiber wastewater is shown in Figure 2a; The changes of $Fe^{2+}$ and $H_2O_2$ concentration in the wastewater after the reaction are shown in Figure 2b,c.

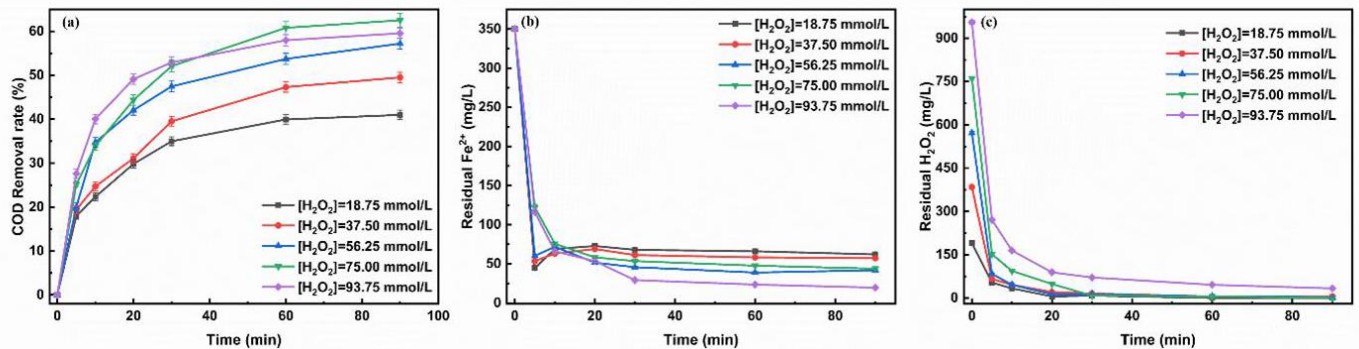

**Figure 2.** Influence of initial concentration of $H_2O_2$ on COD removal rate (**a**), the residual concentration of $Fe^{2+}$ (**b**), and the residual concentration of $H_2O_2$ (**c**).

As shown in Figure 2a, the COD removal efficiency increased from 41.03% to 62.39% as the initial concentration of $H_2O_2$ increased from 18.75 to 75.00 mmol/L. However, when the initial concentration of $H_2O_2$ increased to 93.75 mmol/L, the COD removal rate decreased slightly. The reason is that the increased $H_2O_2$ concentration can accelerate the reaction between $H_2O_2$ and $Fe^{2+}$, leading to the generation of more HO• in the solution, thereby improving the COD removal rate. However, excess $H_2O_2$ will be quenched by HO•, causing the consumption of the HO• and the production of hydrogen peroxide radicals (HOO•) Equation (1) [32]. Simultaneously, the excess $H_2O_2$ will be self-decomposed to produce $H_2O$ and $O_2$, reducing the utilization rate Equation (2).

$$Fe^{2+} + H_2O_2 \rightarrow Fe^{3+} + OH^- + HO\bullet \tag{1}$$

$$H_2O_2 + HO\bullet \rightarrow HOO\bullet + H_2O \tag{2}$$

The higher the concentration of $H_2O_2$ is, the lower the residual concentration of $Fe^{2+}$ will be (Figure 2b). This trend was attributed to the increased $H_2O_2$ concentration promoting the conversion of $Fe^{2+}$ to $Fe^{3+}$. Notably, it can be observed that when the $H_2O_2$ concentration was added to 93.75 mmol/L, the residual amount of $H_2O_2$ in the solution at 30 min was 71.13 mg/L (or 2.09 mmol/L) (Figure 2c). This can be ascribed to the that the low $Fe^{2+}$ concentration is hard to trigger the $H_2O_2$ reaction [33]. Therefore, it can be concluded that the initial concentration of $H_2O_2$ was not as high as possible, and the best initial concentration is 75.00 mmol/L.

### 3.1.2. The Initial Concentration of $Fe^{2+}$

As shown in Figure 3a, the effect of $Fe^{2+}$ concentration on the removal rate of COD at pH 3.0 and $H_2O_2$ concentration of 75 mmol/L. Figure 3b,c shows the changes of $Fe^{2+}$ and $H_2O_2$ concentrations in the aqueous solution after the reaction.

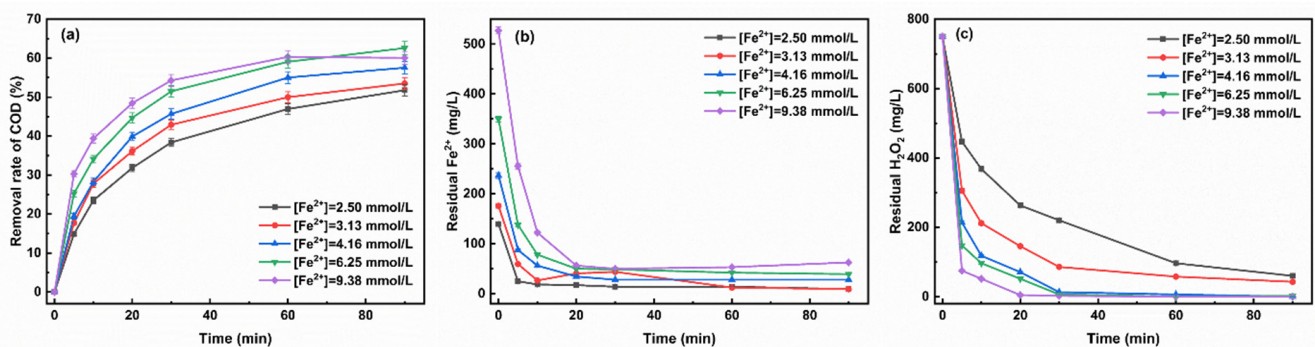

**Figure 3.** Influence of initial concentration of $Fe^{2+}$ on COD removal rate (**a**), the residual concentration of $Fe^{2+}$ (**b**), and the residual concentration of $H_2O_2$ (**c**).

As shown in Figure 3a, with the increase of initial concentration of $Fe^{2+}$, the COD removal rate was increased gradually. The COD removal rate increased significantly from 51% to 65% when the initial concentration of $Fe^{2+}$ increased from 2.50 mmol/L to 6.25 mmol/L. It can be observed that when the initial concentration of $Fe^{2+}$ increases to 9.38 mmol/L, the COD removal rate decreases slightly. The trend of changes in the concentration of $Fe^{2+}$ and $H_2O_2$ after the Fenton reaction was shown in Figure 3b,c. Within the reaction time of 0~5 min (Figure 3b), it was clearly observed that the $Fe^{2+}$ concentration decreased rapidly, and then the curve gradually stabilizes. By comparing the slope of the reaction curve, it can be proved that the increased $Fe^{2+}$ concentration can facilitate the reaction rate. Furthermore, the consumption rate of $H_2O_2$ increased with the increase of the initial concentration of $Fe^2$ $^+$ (Figure 3c). Typically, at a low initial concentration of $Fe^{2+}$, the rate of the reaction Equation (1) was slow and the amount of HO• produced was less than sufficient to oxidize the organic matter in water [17]. Meanwhile, the excess $H_2O_2$ reacted with the generated HO• Equation (2), which consumes $H_2O_2$ in the water that had not been catalytically decomposed, resulting in a lower COD removal rate [34]. However, when the initial concentration of $Fe^{2+}$ was too high, too much HO• was produced at the beginning of the reaction. At the same time, a side reaction Equation (3) occurred, consuming HO• that had not yet participated in the oxidation reaction [35]. In addition, a large amount of HO• would react with each other to form $H_2O$ and $O_2$ Equation (4) [35], which leaded to a decrease in COD removal rate. At a dose of 6.25 mmol/L of $Fe^{2+}$, the $H_2O_2$ reacted completely within 30 min and the best COD removal was achieved. Therefore, the optimal initial concentration of $Fe^{2+}$ was 6.25 mmol/L.

$$Fe^{2+} + HO• \rightarrow Fe^{3+} + OH^- \tag{3}$$

$$4HO• \rightarrow 2H_2O + O_2 \tag{4}$$

### 3.1.3. Initial pH

The effect of initial pH on the COD removal rate of acrylic fiber wastewater is shown in Figure 4a. The change curve of the concentration of $Fe^{2+}$ and $H_2O_2$ after the oxidation-reduction reaction is shown in Figure 4b,c. the initial concentration of $Fe^{2+}$ was 6.25 mmol/L, and the initial concentration of $H_2O_2$ was 75 mmol/L.

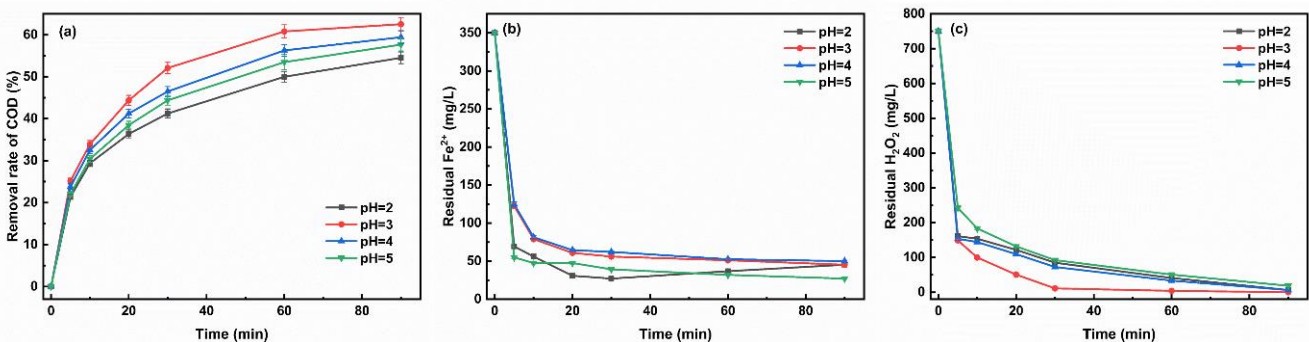

**Figure 4.** Influence of initial pH on COD removal rate (**a**), the residual concentration of $Fe^{2+}$ (**b**), and the residual concentration of $H_2O_2$ (**c**).

The results presented in Figure 4a showed that it can be obtained the highest COD removal rate of 65.79% at a pH of 3.0, then the removal rate decreased slightly as the pH increased. According to the reaction mechanism of the Fenton reagent, the initial pH could directly interfere the complex equilibrium system of $Fe^{2+}$ and $Fe^{3+}$ in the solution, thereby affecting the oxidation ability of the Fenton reagent. As exhibited in Figure 4b,c, when the pH value was 2, the concentration of $Fe^{2+}$ in the solution was relatively low while the concentration of $H_2O_2$ was relatively high within 30 min before the reaction. Due to the higher concentration of $[H^+]$, the reaction Equation (5) was in a reverse reaction state. $Fe^{3+}$ reduction to $Fe^{2+}$ was inhibited in this state, and the catalytic reaction could not proceed smoothly [36]. As a result, the generation rate of $HO\bullet$ was slowed down, and the oxidation capacity was reduced. When the pH value was > 3, it was obvious that the removal rate of COD gradually decreased, mainly due to the increase of $OH^-$ concentration, which leads to the decrease of reaction rate Equation (1), and thus the amount of $HO\bullet$ decreases. As shown in Figure 4b,c, most of the $Fe^{2+}$ and $H_2O_2$ in the system reacted quickly and were consumed during the first 10 min. When the reaction time was between 10 and 30 min, the $Fe^{2+}$ concentration gradually decreased and tended to be stable. When the pH was 2, 4, and 5, the concentration of $H_2O_2$ did not stabilize until 90 min. Notably, the highest utilization and reaction rate of $H_2O_2$ was achieved at pH 3, and the reaction equilibrium could be reached within 30 min. Therefore, the optimal pH value of this study was 3.0, which was consistent with the results of other research.

$$Fe^{3+} + H_2O_2 \rightarrow Fe^{2+} + H^+ + HOO\bullet \tag{5}$$

*3.2. Response Surface Analysis*

3.2.1. Regression Model and Analysis of Variance

The results of the RSM model were presented in Table 3, giving the analysis of variance (ANOVA). It should be noted that the F value (Fisher variation ratio) and *p* value (probability value) in the analysis of the variance table were the leading indicators, showing the significance and adequacy of the model used. *p* value less than 0.05 means that the model was significant, while a value greater than 0.10 was usually regarded as a less critical factor [37]. The ANOVA of this model showed that the F value of lack of fit was 99.01, and the very low *p* value of lack of fit was 0.0003 (<0.05), indicating that these parameters in the model were highly significant. The F value of the regression model was 14.49, and *p* < 0.05, which also proved that the regression model was highly significant, and the experiment was reliable. As can be seen from Table 3, the R-Squared ($R^2$) and Adjusted R-Squared ($R_{aj}^2$) of the model, respectively, were 0.95 and 0.89, indicating that the regression equation was highly reliable, which can explain 89% of the response value changes. Since *p* value was less than 0.05, A, B, $B^2$, $C^2$ can be regarded as vital terms. In addition, it was clear that C, A × B, and A × C were considered irrelevant terms. Therefore, the author obtained the

equation relationship between the response and the variable, which was represented by the second-order polynomial equation fitting based on the coding factors (A, B, C):

$$R = 57.33 - 3.88\,A + 2.86\,B + 0.42\,C - 0.29\,A \times B - 0.85\,A \times C - 2.87\,B \times C - 2.53\,A^2 - 5.76\,B^2 - 10.50\,C^2 \quad (6)$$

where R is the removal of COD, A is the coded value of $H_2O_2$ concentration, B is the coded value of $Fe^{2+}$ concentration, C is the coded value for pH.

**Table 3.** Covariance analysis of linear, quadratic and interaction variables of the regression model.

| Source | Sum of Squares (SS) | Degree of Freedom (df) | Mean Square (MS) | F-Value (F) | *p*-Value (*p*) |
|---|---|---|---|---|---|
| Model | 906.22 | 9 | 100.69 | 14.94 | 0.0009 |
| A-[$H_2O_2$] | 120.51 | 1 | 120.51 | 17.89 | 0.0039 |
| B-[$Fe^{2+}$] | 65.27 | 1 | 65.27 | 9.69 | 0.0170 |
| C-pH | 1.39 | 1 | 1.39 | 0.21 | 0.6629 |
| A × B | 0.33 | 1 | 0.33 | 0.049 | 0.8310 |
| A × C | 2.89 | 1 | 2.89 | 0.43 | 0.5334 |
| B × C | 33.06 | 1 | 33.06 | 4.91 | 0.0623 |
| $A^2$ | 26.94 | 1 | 26.94 | 4.00 | 0.0857 |
| $B^2$ | 139.90 | 1 | 139.90 | 20.76 | 0.0026 |
| $C^2$ | 464.37 | 1 | 464.37 | 68.92 | <0.0001 |
| Residual | 47.16 | 7 | 6.74 | | |
| Lack of Fit | 46.54 | 3 | 15.51 | 99.01 | 0.0003 |
| Pure Error | 0.63 | 4 | 0.16 | | |
| Cor Total | 953.38 | 16 | | | |
| $R^2$ | 0.95 | | | | |
| $R_{aj}^2$ | 0.89 | | | | |

The above regression equation can be used to predict the R-value within the range of the factors in this study. The value and sign of the regression coefficient indicated the influence of each item on the response. It can be seen from Equation (6) that the item with the highest regression coefficient in the interaction term was B*C, which indicated that the interaction term occupies a dominant position in the overall response. The positive sign of the coefficient indicated a synergistic effect, and the negative sign indicated an antagonistic effect.

### 3.2.2. Response Surface Optimization

According to Table 2, the *p* value of A × B, A × C, and B × C were 0.831, 0.533, and 0.0623, respectively. To further determine the interaction between variables, the model Equation (1) was used for fitting, and the result was shown in Figure 5. It was clear that the removal rate of COD increased from 29.20% to 59.30% with the changes of A and B, when the pH was 3 (code value was 0). A view of Figure 5a,d, at the position of the center point, the range of A was near the −0.61 level and the range of B was near the 0.25 level. From Figure 5b,e, at the position of the center point, the range of A was near the −0.65 level and the range of pH was near the 0.10 level. As described in Figure 5c,f, at the position of the center point, the range of B was near the 0.25 level and the range of C was near the 0.1 level. The larger the difference between the coded value of the Center Point and the 0.00 level, the greater the interaction between the two factors [38]. As explained previously, the order of the influence of each variable on COD degradation was: A-[$H_2O_2$] > B-[$Fe^{2+}$] > C-pH. In addition, it was apparent that the interaction between B-[$Fe^{2+}$] and C-pH had a more substantial impact.

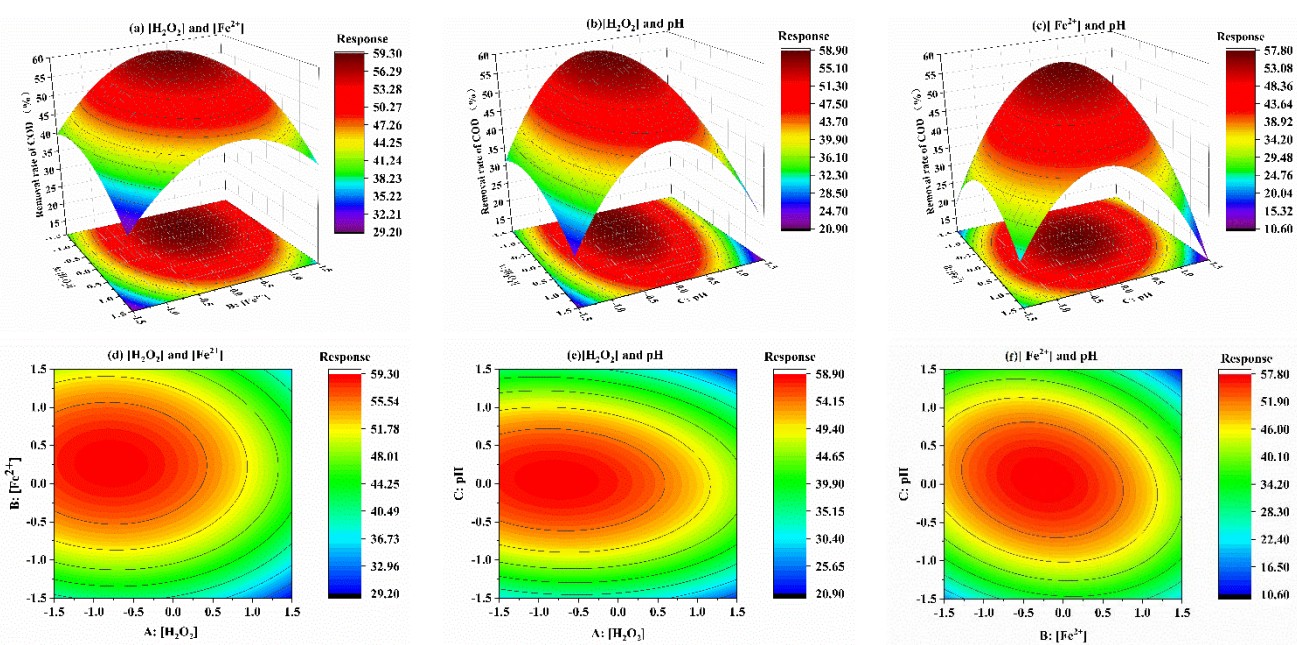

**Figure 5.** A three-dimensional response surface for the response of COD removal rate in terms of coded A and B (**a**), coded A and c (**b**), coded B and C (**c**); The corresponding contour plots for the response of COD removal rate in terms of coded A and B (**d**), coded A and c (**e**), coded B and C (**f**).

### 3.2.3. Verification of Optimal COD Degradation Conditions

As shown in Figure 6, after linear fitting between the predicted and actual values, the $R^2$ was 0.95, which showed that the predicted result had a high degree of credibility [39]. With the use of the Fenton process to degrade COD in acrylic fiber wastewater, and through response surface optimization analysis, the optimal degradation conditions were: (i)the initial concentration of $H_2O_2$ was 60.30 mmol/L; (ii) the initial concentration of $Fe^{2+}$ was 7.46 mmol/L; (iii) the pH was 3; (iv) COD removal rate was 59.22%. The experiment was repeated three times under the above-optimized conditions, and the results were presented in Table 4. Obviously, it can be observed that after treatment, the acrylic fiber wastewater effluent can meet the secondary discharge requirements of the "Integrated Wastewater Discharge Standard" (GB4287-2012) [40].). The COD removal rate's actual and predicted values were 63.2% and 59.8%, and the error was less than 5%. Therefore, the confidence of the obtained model was strong.

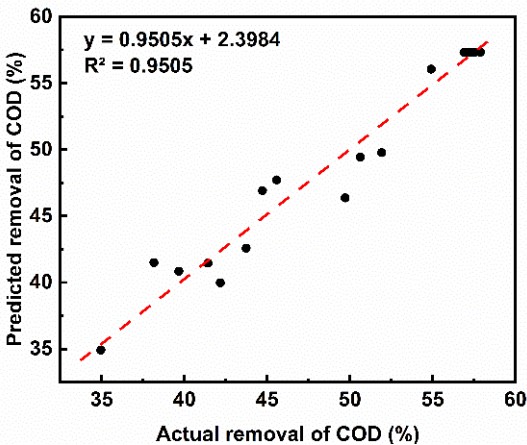

**Figure 6.** The removal of COD: Comparison of actual value and predicted value.

**Table 4.** Removing effect of acrylic fiber wastewater by Fenton process.

| Water Quality Index | Influent | Effluent | Emission Limit |
|---|---|---|---|
| pH | 5.5 ± 0.2 | 6.4 ± 0.2 | 6~9 |
| COD/(mg·L$^{-1}$) | 259.6 ± 8.5 | 93.3 ± 3.5 | 200 |
| TOC/(mg·L$^{-1}$) | 92.3 ± 2.5 | 28.2 ± 0.7 | 30 |
| NH$_4^+$-N/(mg·L$^{-1}$) | 63.5 ± 2.6 | 26.4 ± 0.6 | 25 |
| TN/(mg·L$^{-1}$) | 83.8 ± 3.2 | 34.5 ± 0.6 | 50 |

*3.3. Removal Effect of Refractory Organic Matter*

In order to evaluate the degradation effect of toxic and difficult-to-degrade organic pollutants in acrylic fiber wastewater, gas chromatography-mass spectrometry (GC-MS) was used to detect toxic and hazardous substances in the wastewater. Based on the optimal process conditions (The initial concentration of H$_2$O$_2$ was 60.30 mmol/L, the initial concentration of Fe$^{2+}$ was 7.46 mmol/L, the pH was 3), the actual acrylic manufacturing wastewater was treated by the Fenton method. The results were presented in Figure 7. It was evident that the chromatographic peaks of most organics in the water after Fenton treatment were weakened. Moreover, the raw water contained 16 primary organic pollutants, mainly including aromatic hydrocarbons and long-chain alkanes. It can be seen that the concentration of organic pollutants had been significantly reduced after the Fenton process, demonstrating that the Fenton process can convert refractory organic matter into small molecular organics or inorganics [41]. In addition, as shown in Table 5, Undecane, Diisobutyl phthalate and Dibutyl phthalate still existed in the effluent. We speculated that Undecane might be the final product of the degradation of long-chain alkanes. In the Fenton oxidation process, HO● preferentially oxidized small molecular organic compounds, followed by other large molecular organic compounds [42]. Since Diisobutyl phthalate and Dibutyl phthalate were macromolecular organic compounds, and the concentration in the influent was relatively high. As a result, Diisobutyl phthalate and Dibutyl phthalate in the effluent was not wholly removed. Notably, the Fenton process can remove most of the refractory organic matter in acrylic fiber wastewater.

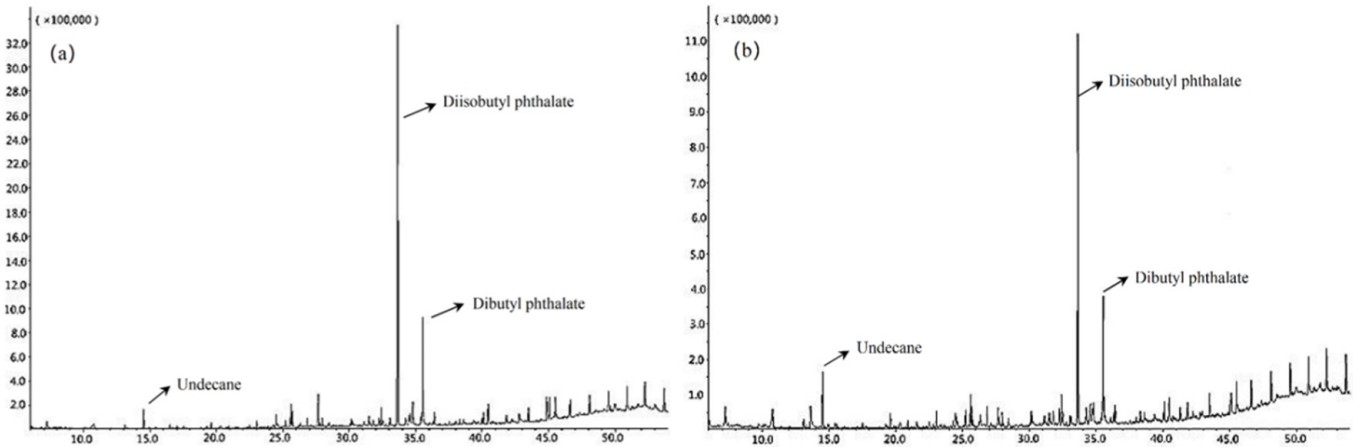

**Figure 7.** GC-MS patterns: (**a**) raw water, (**b**) water samples treated by Fenton process.

**Table 5.** The main organic pollutants.

| Number | Peak Time (min) | Organic Matter | Structural Formula | Molecular Formula | The Degree of Match (%) | Occasion Influent | Occasion Effluent |
|--------|-----------------|----------------|--------------------|-------------------|-------------------------|-------------------|-------------------|
| 1 | 14.532 | Undecane | | $C_{11}H_{24}$ | 94 | √ | √ |
| 2 | 24.509 | 1,4,5-Trimethylnaphthalene | | $C_{13}H_{14}$ | 62 | √ | |
| 3 | 25.643 | 2,6-Di-tert-butyl-4-methylphenol | | $C_{15}H_{24}O$ | 96 | √ | |
| 4 | 25.732 | 2,4-di- tert -butylphenol | | $C_{14}H_{22}O$ | 94 | √ | |
| 5 | 27.657 | 2,2,4-trimethyl-1,3-pentanediol diisobutyrate | | $C_{16}H_{30}O_4$ | 59 | √ | |
| 6 | 32.404 | Octadecane | | $C_{18}H_{38}$ | 99 | √ | |
| 7 | 33.656 | Diisobutyl phthalate | | $C_{16}H_{22}O_4$ | 90 | √ | √ |
| 8 | 35.535 | Dibutyl phthalate | | $C_{16}H_{22}O_4$ | 94 | √ | √ |
| 9 | 40.446 | 2,6-diphenylpyridine | | $C_{17}H_{13}N$ | 97 | √ | |
| 10 | 43.488 | Tetracosane | | $C_{24}H_{50}$ | 98 | √ | |
| 11 | 44.882 | Benzo(H)quinoline | | $C_{15}H_{13}N$ | 50 | √ | |
| 12 | 45.073 | Heneicosane | | $C_{21}H_{44}$ | 93 | √ | |
| 13 | 45.51 | DEHP, di-(2-ethylhexyl) phthalate. | | $C_{16}H_{22}O_4$ | 90 | √ | |
| 14 | 46.617 48.100 49.528 | Hexacosane | | $C_{26}H_{54}$ | 98 | √ | |
| 15 | 50.907 53.703 | Octacosane | | $C_{28}H_{58}$ | 98 | √ | |
| 16 | 52.240 | Eicosane | | $C_{20}H_{42}$ | 98 | √ | |

### 3.4. Mechanism Analysis

In brief, the degradation of acrylic fiber wastewater by the Fenton process was based on the HO•, which can degrade organic pollutants through oxidation, generated during the electron transfer process between $H_2O_2$ and a homogeneous metal catalyst ($Fe^{2+}$) [43]. Equations (1)–(5), (7)–(9) described the essential reactions in the Fenton process [2,44–47]. The production of HO• represented the beginning of the process, which the interaction of

$Fe^{2+}$ with $H_2O_2$ Equation (1). Meanwhile, the generated $Fe^{3+}$, which were reduced to $Fe^{2+}$ by $H_2O_2$, undergo a Fenton-like reaction Equation (5). When there were too many $Fe^{2+}$, it would react with HO• to produce $Fe^{3+}$ Equation (3). Therefore, Equations (1) and (3) were the primary sources of $Fe^{3+}$. Excessive $H_2O_2$ reacted with HO• to generate HOO•, which oxidation activity was lower than HO•.

A large amount of HOO• was consumed through reacting with $Fe^{3+}$, $Fe^{2+}$, and HO• Equations (7)–(9), enabling an effective cyclic mechanism of $Fe^{3+}$ and $Fe^{2+}$.

$$Fe^{3+} + HOO• \rightarrow Fe^{2+} + H^+ + O_2 \tag{7}$$

$$Fe^{2+} + HOO• \rightarrow Fe^{3+} + HO_2^- \tag{8}$$

$$HOO• + HO• \rightarrow H_2O + O_2 \tag{9}$$

In addition, a great deal of HO•, which had high oxidation, was produced through the above reaction. In order to demonstrate that the degradation of acrylic acid wastewater is based on the oxidation of HO•, HO• scavenging experiments based on the degradation of acrylic acid wastewater by Fenton system were carried out. It has been shown that isopropyl alcohol reacts rapidly with HO- and competes with the target substrate, thus effectively inhibiting the reaction of the target substrate with HO• [48]. Therefore, in this experiment isopropyl alcohol (IPA) was used to remove the active species associated with the degradation of organic matter. Figure 8 shows the effect of using different concentrations (0%, 2%, 6%, 10%) of isopropanol on the degradation of organic matter in acrylic acid wastewater based on the optimal fenton process conditions (mentioned in Section 3.2.3). As can be seen from Figure 8, the inhibition effect of isopropanol is obvious, with 2%, 6%, and 10% isopropanol almost completely inhibiting the degradation of organics. It indicates that HO• is the main active species in this Fenton system for the degradation of organic matter. Therefore, the mechanism in the degradation process of organic matter in acrylic wastewater was speculated as follows. In the presence of organic molecules (RH) from acrylic manufacturing wastewater, the generated HO• attacked the organic molecules by absorbing protons, producing highly reactive organic radicals (R•), which could be further oxidized to generate ROH Equations (10) and (11) [20,43,49]. Further, ROH was oxidized by HO• to form $R_1OOH$ and $R_2H$ to achieve the conversion of organic macromolecules to organic small molecules Equations (10) and (11). In addition, $R_1OOH$ was easily oxidized by HO• to form smaller molecular weight organic matter ($R_3H$), $H_2O$ and $CO_2$ Equation (12). The efficient degradation of organic matter in acrylic wastewater was achieved through the oxidation reactions of Equations (10)–(13) (where R, $R_1$, $R_2$, and $R_3$ represent one or more products in the degradation stage). Furthermore, $H_2O_2$ was generated via a reaction among HO• Equation (14).

$$HO• + RH \rightarrow R• + H_2O \tag{10}$$

$$HO• + R• \rightarrow ROH \tag{11}$$

$$HO• + ROH \rightarrow R_1OOH + R_2H \tag{12}$$

$$HO• + R_1OOH \rightarrow R_3H + CO_2 + H_2O \tag{13}$$

$$HO• + HO• \rightarrow H_2O_2 \tag{14}$$

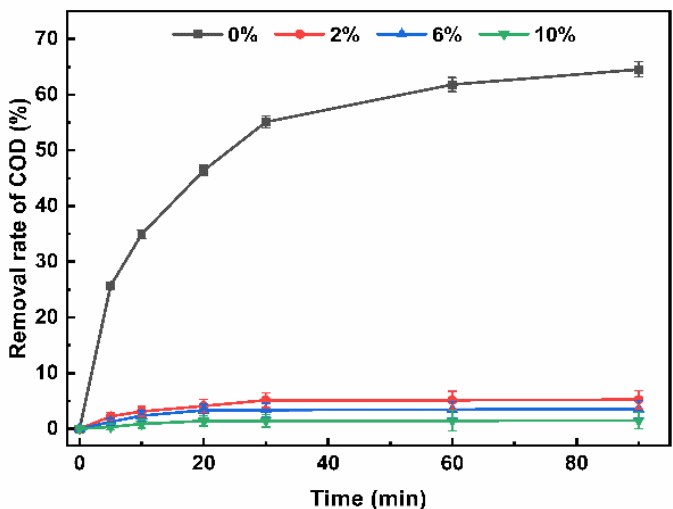

**Figure 8.** Effect of HO• scavenger on the degradation of organic matter in acrylic acid wastewater.

The mechanism for the degradation of acrylic manufacturing wastewater by the Fenton process is explained in Figure 9. The chain reaction can continue until the $H_2O_2$ was completely consumed because $Fe^{2+}$ acted as a catalyst and transmitter in the reaction. Various free radicals and intermediate substances, such as HOO•, $Fe^{3+}$ and HO•, were generated during the reaction as the nodes of the chemical reaction chain. In addition, $Fe^{3+}$ was readily hydrolyzed and forms complexes. The flocculation and precipitation processes of these complexes also played an important role in removing organic matter. The degradation of organic matter was achieved through a series of reactions as described above.

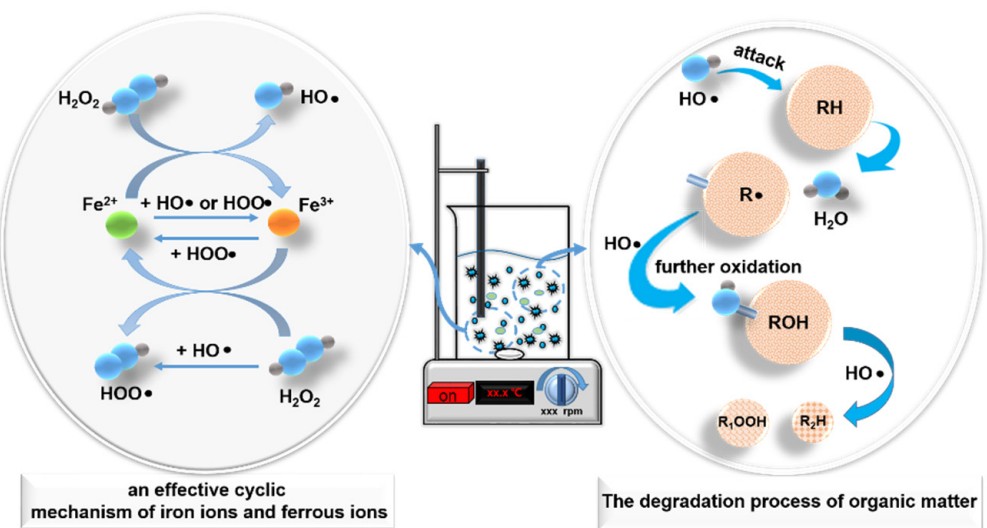

**Figure 9.** Schematic diagram of Fenton reaction process.

## 4. Conclusions

(1) The optimization analysis combined with the response surface method showed that the optimal degradation conditions for acrylic fiber wastewater using the Fenton method are: (i) initial $H_2O_2$ concentration of 60.90 mmol/L; (ii) initial $Fe^{2+}$ concentration of 7.44 mmol/L; (iii) pH of 3. The predicted degradation efficiency of the model equation was 59.8%, and the actual COD degradation rate was 63.2%.

(2) The deviation between the actual and fitted model values was less than 5%, indicating that the model equation had a high degree of credibility. According to the analysis of the influence of factors and variables, it can be seen that the influence order of the

three factors was $[H_2O_2] > [Fe^{2+}] > pH$. In addition, the interaction between $[Fe^{2+}]$ and pH had the most significant impact on the degradation of COD.

(3) For the actual acrylic fiber wastewater treatment, the removal rate of COD, TOC, $NH_4^+$-N, TN is 61.45%~66.51%, 67.82%~70.99%, 55.67%~60.97%, 56.45%~61.03%, respectively. The effluent met the textile dyeing and finishing industry water pollutant discharge standard "GB4287-2012". The COD, TOC, $NH_4^+$-N, and TN were decreased to $93.3 \pm 3.5$ mg/L, $28.2 \pm 0.7$ mg/L, $26.4 \pm 0.6$ mg/L, $34.5 \pm 0.6$ mg/L, respectively.

(4) HO• generated during electron transfer between $H_2O_2$ and $Fe^{2+}$ effectively decompose organic pollutants in acrylic production wastewater. In this case, 13 kinds of aromatic hydrocarbons and long-chain alkanes in acrylic fiber wastewater had been effectively removed.

**Author Contributions:** Conceptualization, Z.L. and C.Z.; methodology, Z.L.; software, Z.L., and P.S.; validation, Z.L., C.Z. and P.S.; formal analysis, Z.L.; investigation, Z.L.; resources, Z.L.; data curation, Z.L.; writing—original draft preparation, Z.L.; writing—review and editing, P.S. and Z.L.; visualization, W.L. and Z.Z.; supervision, Z.Z., X.W. and W.H.; project administration, C.Z.; funding acquisition, C.Z. All authors have read and agreed to the published version of the manuscript.

**Funding:** This research received no external funding.

**Institutional Review Board Statement:** "Not applicable" for studies not involving humans or animals.

**Informed Consent Statement:** "Not applicable." for studies not involving humans.

**Data Availability Statement:** Data are contained within the article.

**Acknowledgments:** This research is supported by the National Research Foundation China, and China University of Mining and Technology (Beijing) under the laboratory of the Institute for Total Process Pollution Control and Circular Economy.

**Conflicts of Interest:** The authors declare no conflict of interest.

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
