# Peer review of "Fenton Process for Treating Acrylic Manufacturing Wastewater: Parameter Optimization, Performance Evaluation, Degradation Mechanism"

_water, doi:10.3390/w14182913_

Round 1

Reviewer 1 Report

The submitted manuscript (ID: water-1898660) exploited Fenton technology to treat acrylic fiber manufacturing wastewater. The impacts of key operating variables including the initial concentration of H2O2, the initial concentration of Fe2+, and solution pH on the COD removal rate (RCOD) were explored and the treatment process was optimized by Response Surface Methodology (RSM), but the content of the manuscript is too tedious and unreadable. In addition, there are still some issues that need to be clarified before the manuscript can be considered further.

1.      Why use concentrated sulfuric acid to pretreat acrylic wastewater with a pH of 5.4~5.8d before sampling?

2.      In the experimental methods section, since it has been described that there are many factors that affect the degradation of COD by the Fenton method, why only choose the initial concentration of H2O2, the initial concentration of Fe2+, and the pH value as the main influencing factors in the following text? please explain why these three factors are the main factors.

3.      Figures 2, 3, and 4 are extremely unclear. It is recommended to adjust the picture layout and font size.

4.      Diisobutyl phthalate and Dibutyl phthalate in the water samples treated by the Fenton method are less than that in the raw water, how can it be proved that they are the final degradation products of long-chain alkanes?

5.      Can experiments be designed to demonstrate that acrylic acid wastewater degradation is based on hydroxyl radicals by using free radical scavengers?

6.      Compared with the previous Fenton method to degrade acrylic wastewater, what are the highlights of this article? Or what new mechanism has been discovered?

7.      How the mechanism described by the authors corresponds to the high degradation rate of acrylic wastewater?

Reviewer 2 Report

In this manuscript, Lin et al. discussed the parameter optimization, performance evaluation, and degradation mechanism of the Fenton process for treating acrylic manufacturing wastewater. Overall, this work was well organized and presented and appeals to the readership of Water. This reviewer would like to support its publication after the below minor points are properly addressed.

1. Figure 6, the slope of the fitting and the R2 were the same number (0.9505). Please double check.

2. Figure 7, Chinese letters are seen in the figure. Please revise.

3. Equations 8 and 12 were not balanced.

4. How was the initial pH controlled? More experimental details should be provided.

5. The detection of ammonia nitrogen, Fe2+, H2O2 was only briefly mentioned. More details about the respective detection methods should be provided in the Experimental section.

6. Very recent works on water treatment are suggested to be referenced in the Introduction (e.g., ACS Sustainable Chem. Eng. 2022, 10, 1899-1909).

7. The writing can be improved. For example, line 104, the sentence “and adjusted the pH of the 104 solution with 1.0 mol/L H2SO4 (or NaOH) solution” need to be revised.

Author Response

Response to Reviewer 2 Comments

Point 1: Figure 6, the slope of the fitting and the R2 were the same number (0.9505). Please double check.

Response 1: Upon verification and inspection, the fitted slope values are 0.9505 and R2 is 0.9505 in Figure 6. It is purely by chance that the fitted slope values are the same as the R2 values. Thanks for your valuable counsel. The specific data are as follows.

Run

S1

S2

S3

S4

S5

S6

S7

S8

S9

S10

S11

S12

S13

S14

S15

S16

S17

Actual COD removal rate

51.92

43.73

54.91

45.57

44.72

39.66

50.63

42.17

34.95

49.71

38.16

41.42

57.88

57.28

57.04

57.54

56.89

Predicted COD removal rate

49.77

42.58

56.06

47.72

46.91

40.85

49.44

39.98

34.91

46.37

41.50

41.46

57.33

57.33

57.33

57.33

57.33

(Note: The operating conditions of S1~17 are consistent with subsection 2.3 in the article.)

Point 2: Figure 7, Chinese letters are seen in the figure. Please revise.

Response 2: I am very sorry for this error, it has been corrected. Thanks for your valuable counsel. 

Point 3: Equations 8 and 12 were not balanced.

Response 3: Thanks for your valuable counsel. After checking, Eq. 8 was found to be written incorrectly. Eq. 8 should be expressed as follows:Fe2+ + HOO• → Fe3+ + HO2-.

We sorry for the lack of detailed explanation for Eq. 12. R, R1, R2, in Eq. 12 (HO• + ROH → R1OOH+R2H) denoted organic compounds with different molecular chains, respectively. R1OOH and R2H were the products of the oxidative decomposition of ROH by HO•.

Point 4: How was the initial pH controlled? More experimental details should be provided.

Response 4: The pH adjustment was achieved by titrating 0.1 or 1 mol/L of H2SO4 (or NaOH) solution under the condition of stirring intensity around 140 rpm. In which the pH of the solution was monitored by (LEICI PHB-4) portable pH meter to achieve precise regulation of the solution pH. Experimental details were added in lines 113-117 of the article. Thanks for your valuable counsel.

Point 5: The detection of ammonia nitrogen, Fe2+, H2O2 was only briefly mentioned. More details about the respective detection methods should be provided in the Experimental section.

Response 5: Thanks for your valuable counsel. Lines 125 to 140 of the article are supplemented with detailed descriptions of the detection methods for COD, ammonia nitrogen, Fe2+, Fe3+, and H2O2.

More details are described below:

The method of COD detection refers to the standard method "Water quality-Determination of the chemical oxygen demand-Dichromate method, HJ 828-2017" published by the Ministry of Ecology and Environment of the People's Republic of China [1]. 10 ml of water sample was placed in a conical flask. 0-2 ml of mercury sulfate solution (100 g/L), 5.00 ml of potassium dichromate standard solution (0.25 or 0.025 mol/L) and 2-3 grains of antiboiling glass were added. From the upper end of the condenser tube, 15 ml of silver sulfate-sulfuric acid solution was slowly added to prevent the escape of low boiling point organics. The solution was kept at microboiling reflux for 2 h. After refluxing and cooling, the conical flask was removed. The solution was cooled to room temperature and 3 drops of test ferrous spirit indicator solution were added. The solution was titrated with ferrous ammonium sulfate standard solution (0.05 or 0.005 mol/L), and the color of the solution changed from yellow to red-brown through blue-green.

The detection method of ammonia nitrogen refers to the standard method "Water quali-ty-Determination of ammonia nitrogen-Nessler's reagent spectrophotometry, HJ535-2009" published by the Ministry of Ecology and Environment of the People's Republic of China [2]. We placed 50 ml of water sample in a 50 ml cuvette and added 1.0 ml of potassium sodium tartrate solution. Then 1.5ml of nano reagent was added and shaken well. The absorbance was measured with a 20mm cuvette at a wavelength of 420nm after being left for 10min.

The determination method of iron content refers to the standard method "Water quality-Determination of Iron-phenanthroline spectrophotometry, HJ/T 345-2007" published by the Ministry of Ecology and Environment of the People's Republic of China [3]. (i) Total iron was determined as follows. We took the sample and immediately acidified it with hydrochloric acid to pH < 1. 50.0 mL of water sample was placed in a 150 mL conical flask, followed by the addition of 1 mL of (1+3) hydrochloric acid and 1 mL of hydroxylamine hydrochloride solution. The sample was heated and boiled until the volume is reduced to about 15 mL to ensure the dissolution and reduction of all iron. A piece of Congo red test paper was added, followed by the addition of saturated sodium acetate solution dropwise until the test paper just turned red. 5mL of buffer solution, 2mL of 0.5% o-phenanthroline solution was added. Then we added deionized water to 50ml. After 15 min, the absorbance at 510 nm was measured with a 10 mm cuvette, using water as a reference. (ii) Ferrous ions were determined as follows. The 15 ml water sample was filtered and placed in a 50 ml colorimetric tube. 5mL of buffer solution, 2mL of 0.5% o-phenanthroline solution was added. Then we added deionized water to 50ml. After 15 min, the absorbance at 510 nm was measured with a 10 mm cuvette, using water as a reference.

The concentration of H2O2 was determined spectrophotometrically with potassium titanium oxalate. 5 ml of water sample was added to a 25 ml cuvette. 5 ml of potassium titanium oxalate solution (0.02 mol/L) was added. Deionized water was used to dilute to the scale. After standing for 8 min, the absorbance was determined at 385 nm. Deionized water was used as a reference during the determination. [4,5].

Point 6: Very recent works on water treatment are suggested to be referenced in the Introduction (e.g., ACS Sustainable Chem. Eng. 2022, 10, 1899-1909).

Response 6: It is a great pleasure to read your recommended articles. The citation description of the article is shown in lines 57 to 60. ” X. Xu and Z. Shao et al. [6]developed highly efficient peroxy monosulfate activated catalysts (e.g. LaSrCo0.8Fe0.2O4) that showed excellent performance in catalytic advanced oxidation applications of difficult to degrade organic pollutants.”

Point 7: The writing can be improved. For example, line 104, the sentence “and adjusted the pH of the solution with 1.0 mol/L H2SO4 (or NaOH) solution” need to be revised.

Response 7: In lines 113~117 of the article, the revision has been completed. Specifically modified to " The pH adjustment was achieved by titrating 0.1 or 1 mol/L of H2SO4 (or NaOH) solution under the condition of stirring intensity around 140 rpm.". Thanks for your valuable counsel.

  1. NEPA. Water quality-Determination of the chemical oxygen demand-Dichromate method. China National Standard HJ828-2017. China National Environmental Protection Administration(NEPA). 2017. https://www.mee.gov.cn/ywgz/fgbz/bz/bzwb/jcffbz/201704/W020170606398873416325.pdf.
  2. NEPA. Water quality-Determination of ammonia nitrogen-Nessler's reagent spectrophotometry. China National Standard HJ535-2009. China National Environmental Protection Administration(NEPA). 2009. https://www.mee.gov.cn/ywgz/fgbz/bz/bzwb/jcffbz/201001/W020180319535685015821.pdf.
  3. NEPA. Water quality—Determination of Iron—phenanthroline spectrophotometry. China National Standard HJ/T 345─ 2007. China National Environmental Protection Administration(NEPA). 2007. https://www.mee.gov.cn/ywgz/fgbz/bz/bzwb/jcffbz/200703/W020120104561152753262.pdf.
  4. Zhao, H.; Dong, M.; Wang, Z.; Wang, H.; Qi, H. Roles of free radicals in NO oxidation by Fenton system and the enhancement on NO oxidation and H2O2 utilization efficiency. Environmental Technology 2020, 41, 109-116. https://doi.org/10.1080/09593330.2018.1491638.
  5. LU, P. Spectrophotometric determination of hydrogen peroxide in Fenton advanced oxidation systems by potassium titanyl oxalate. Architectural Engineering Technology and Design 2014, 582-582,517. https://doi.org/10.3969/j.issn.2095-6630.2014.08.553.
  6. Yang, L.; Jiao, Y.; Xu, X.; Pan, Y.; Su, C.; Duan, X.; Sun, H.; Liu, S.; Wang, S.; Shao, Z. Superstructures with Atomic-Level Arranged Perovskite and Oxide Layers for Advanced Oxidation with an Enhanced Non-Free Radical Pathway. ACS Sustainable Chemistry & Engineering 2022, 10, 1899-1909. https://doi.org/10.1021/acssuschemeng.1c07605.

Round 2

Reviewer 1 Report

The revised manuscript can be accepted for publication.